# Basking shark sub-surface behaviour revealed by animal-towed cameras

Jessica L. Rudd[1], Tiago Bartolomeu[2], Haley R. Dolton[3], Owen M. Exeter[3], Christopher Kerry[3], Lucy A. Hawkes[1]*, Suzanne M. Henderson[4], Marcus Shirley[5], Matthew J. Witt[1,3]

1 Hatherly Laboratories, University of Exeter, College of Life & Environmental Sciences, Exeter, United Kingdom, 2 CEiiA, Matosinhos, Portugal, 3 Environment and Sustainability Institute, University of Exeter, Penryn, United Kingdom, 4 NatureScot, Inverness, United Kingdom, 5 MrROV, Wadebride, Cornwall, United Kingdom

* l.hawkes@exeter.ac.uk

## Abstract

While biologging tags have answered a wealth of ecological questions, the drivers and consequences of movement and activity often remain difficult to ascertain, particularly marine vertebrates which are difficult to observe directly. Basking sharks, the second largest shark species in the world, aggregate in the summer in key foraging sites but despite advances in biologging technologies, little is known about their breeding ecology and sub-surface behaviour. Advances in camera technologies holds potential for filling in these knowledge gaps by providing environmental context and validating behaviours recorded with conventional telemetry. Six basking sharks were tagged at their feeding site in the Sea of Hebrides, Scotland, with towed cameras combined with time-depth recorders and satellite telemetry. Cameras recorded a cumulative 123 hours of video data over an average 64-hour deployment and confirmed the position of the sharks within the water column. Feeding events only occurred within a metre depth and made up ¾ of the time spent swimming near the surface. Sharks maintained similar tail beat frequencies regardless of whether feeding, swimming near the surface or the seabed, where they spent surprisingly up to 88% of daylight hours. This study reported the first complete breaching event and the first sub-surface putative courtship display, with nose-to-tail chasing, parallel swimming as well as the first observation of grouping behaviour near the seabed. Social groups of sharks are thought to be very short term and sporadic, and may play a role in finding breeding partners, particularly in solitary sharks which may use aggregations as an opportunity to breed. In situ observation of basking sharks at their seasonal aggregation site through animal borne cameras revealed unprecedented insight into the social and environmental context of basking shark behaviour which were previously limited to surface observations.

## Introduction

The development of tracking technologies over the past 50 years has revolutionised the field of ecology by collecting minimally biased, continuous data on free-ranging animal movement,

**Data Availability Statement:** Data for this manuscript has been made available on Figshare with the following DOIs. CmaxVideoAnalysis_2018_dutycycled_MW.xlsx

(10.6084/m9.figshare.14837841)
CmaxVideoAnalysis_2018_continuous_MW.xlsx
(10.6084/m9.figshare.14837844)
CmaxVideoAnalysis_2019_dutycycled_MW.xlsx
(10.6084/m9.figshare.14837838).

**Funding:** NatureScot and the University of Exeter
funded the study. NatureScot is an independent
scientific adviser to the Scottish Government. The
funder provided support in the form of salaries for
authors [S.M.H], but did not have any additional
role in the study design, data collection and
analysis, decision to publish, or preparation of the
manuscript. The University of Exeter is a research-
intensive university based in the UK. The specific
roles of authors are articulated in the 'author
contributions' section.

**Competing interests:** Since the funder,
NatureScot, is not a commercial organisation, we
do not have any commercial affiliations. This does
not alter our adherence to PLOS ONE policies on
sharing data and materials.

behaviour, physiology or environment [1–6]. While biologging tags have answered a wealth of ecological questions, the drivers and consequences of movement and activity often remain difficult to ascertain, particularly in the marine environment where mobile species are challenging to observe directly. A similar looking dive profile may reflect several types of behaviour [7], while spatial overlap between animals does not necessarily mean that they interact, particularly when data is recorded at different spatial or temporal scales (i.e. fishing vessel tracking data is recorded at coarser resolution than animal-borne devices [8]). While cameras have been deployed on animals since the early 1900s [9], despite device miniaturisation and advances in unit recovery systems (i.e. radio and tracking technologies), applications in behavioural research remain relatively recent.

The use of animal-borne cameras in tandem with environmental or movement sensors has not only been paramount in validating behaviours recorded by tags on free-ranging animals [10–12], but has also given new understanding of fine-scale habitat selection, which would remain undetected using conventional telemetry systems. For example camera tags have revealed when dive depth may be constrained by bathymetry and not by behaviour [13], recorded previously unknown interactions between white sharks (*Carcharodon carcharias*) in kelp forests [14], and sixgill sharks (*Hexanchus griseus*) associating with rocky outcrops at 700 m depth [11]. Through direct observations of feeding events, animal-borne cameras have also revealed the diet [15–19], feeding rates [20, 21], sensory systems [22, 23] and foraging strategies [7, 12, 24] of a range of marine species. For example, little penguins (*Eudyptula minor*) are more likely to hunt cooperatively when foraging for aggregating prey than when prey are solitary [25], while instances of kleptoparasitism have been recorded by leopard seals (*Hydrurga leptonyx*) [26] and Hawaiian monk seals (*Monachus schauinslandi*) [27], which would otherwise go undetected through tracking data alone. Animal-borne cameras can also document cryptic behaviours [28], the use of tools [29], as well as mating in marine mammals [30–32] and sharks [33], helping to identify possible breeding sites [34].

Basking sharks (*Cetorhinus maximus*) are the world's second largest fish species [35] with a circumglobal distribution in temperate waters [36, 37]. They are susceptible to bycatch [38] and are Endangered (IUCN) throughout their range [39]. The northeast Atlantic is home to several internationally important conspicuous seasonal foraging aggregations, including the Isle of Man, the south west of England, the west of Ireland and the Sea of Hebrides on the west coast of Scotland [40–44]. This last site has been subject to scientific study to improve the evidence base concerning a proposed Marine Protected Area (MPA), specifically intended for the protection of basking sharks, and also minke whales (*Balaenoptera acutorostrata*). Previous research efforts on basking sharks has advanced knowledge on their ecology, including foraging and diving activities [45–47], energetic expenditure [48], habitat preference [49, 50], long distance and seasonal migration patterns [36, 41, 51–53], site fidelity [41, 54, 55], and diel behaviour [56]. Comparatively, little is known of their reproductive biology, or the environmental or social context of sub-surface activity.

While basking sharks are likely to aggregate in the Sea of Hebrides to feed [43, 57], conspicuous behaviours not related to foraging have been observed such as nose-to-tail following, echelon swimming and breaching [43, 57–59], which are thought to be associated with courtship displays as reported in number of elasmobranch species [60–64]. There is no definite knowledge of the time spent by sharks interacting with conspecifics, whether this occurs sub-surface, and if courtship-like behaviour results in breeding attempts [49, 57, 65]. Such information would support previous findings of the importance of the Sea of Hebrides for basking sharks and aid the development of management plans within the proposed MPA. Animal-borne cameras therefore represent a promising tool to explore social interactions in a cryptic environment and were used in the present study to (i) investigate sub-surface behaviour, (ii) record

interactions with conspecifics, (iii) improve knowledge on their fine scale habitat and depth use and (iv) reveal putative threats.

## Materials and methods

### Study site

All work was carried out in accordance with the UK HM Government Home Office under the Animals (Scientific Procedures) Act 1986 (Project Licence P23C6EFD) and under the Wildlife & Countryside Act 1981 (as amended) (Licence: 124812), and were reviewed and approved by the University of Exeter's animal welfare and ethics review board (AWERB). Basking sharks were tagged with camera tags in the vicinity of the islands of Coll and Tiree within the Inner Hebrides, Scotland (N 56˚33', W 6˚41') in August 2018 and July 2019 (Fig 1A and 1B). Sharks were tagged by approaching them from behind using a 10 m vessel, until close enough to apply the camera tag system using a weighted 4 m long Hawaiian sling darting pole. Tags were attached to the base of the primary dorsal fin using a titanium dart (Wildlife Computers WA, USA). Sharks were sexed at the time of tagging using a sub-surface video camera system.

### Camera tags

Towed video camera tags (Fig 1C) comprised of four components: (i) a towed buoyant hydro-dynamic body (CEiiA, Portugal) that encapsulated (ii) a camera system recording data with forward and rearward point-of-view (MrROV, UK), (iii) a G5 time-depth recorder (Cefas, UK) gathering information on temperature and depth at 1 Hz, and (iv) a SPOT6 satellite transmitter (Wildlife Computers, USA) communicating with the Argos System. The camera tags did not have lighting capability, which would have drawn on the battery, reduced the recording duration (instead relying on natural light during the daytime) and may have influenced the behaviour of the tagged animal or conspecifics. The combined package had a total weight of 1.3 kg in air, with a net buoyancy of 0.3 kg. Optimal towing dynamics of the system (e.g. attitude and stability) were derived for a scenario assuming a majority movement velocity of 2 m/sec (derived using electronic tags with integrated speed measurement [59]. Each tag system was attached to a titanium dart via a monofilament tether (1.8 m length in 2018; 1.5 m length in 2019), which comprised of a swivel and either an electronic Payload Release Device (PRD, Wildlife Computers, USA), enabling detachment of the tag from the shark at a pre-specified date and time (n = 5) or a manual galvanic corrosive link (model A2; International Fishing Devices, FL, USA; n = 1). In 2018, the PRDs used on two sharks, were programmed to detach at 06:00 BST the day following attachment and the galvanic release was estimated to provide a 12-hour attachment period given an approximate water temperature of 12˚C. In 2019, PRDs were used on all deployments and programmed to release after a period of four days, which covered a period of forecasted poor weather and as such ensured cameras did not detach during periods when they could not be retrieved safely. Once the PRD activated, the camera tag detached from the dart, floated to the surface and position identified using the Argos System via the onboard SPOT6. Once the retrieval vessel was within 5 km of the most recent Argos location, a hand-held IC-R20 VHF receiver with directional antenna (ICOM, UK) and an RXG-234 goniometer (CLS, Toulouse, France) were used to aid retrieval. Towed cameras were programmed to record video data at 1080p resolution at 50 frames per second until either the memory or power were exhausted. Video recording was continuous in 2018, but a duty cycle of 30 seconds of recording every 5 minutes was used in 2019 to extend the recording duration. The 2018 continuous video data were therefore sub-sampled to 30-second video sequences every 5 minutes to match the duty cycling used in 2019. We also compared habitat use and

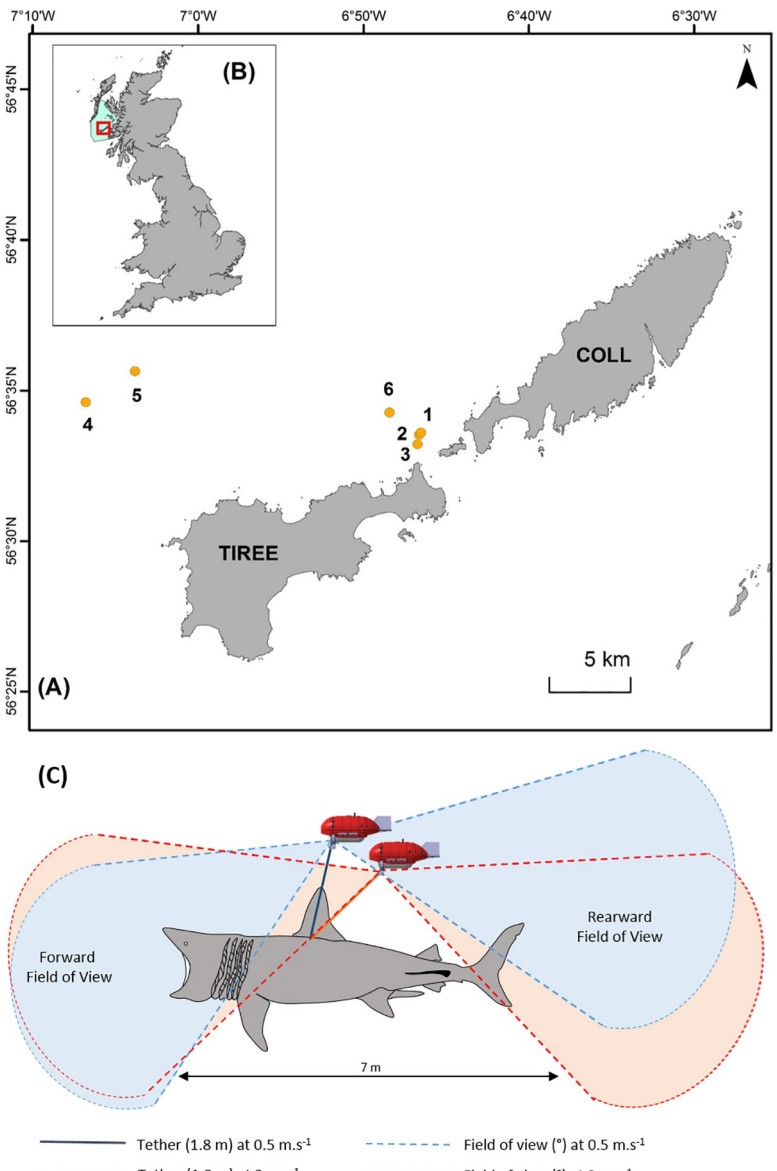

**Fig 1. Study site and towed camera tags.** (**A**) Study site and tagging location of the six basking sharks (orange circles) within the Sea of the Hebrides Marine Protected Area (blue polygon) (**B**). (**C**) Towed camera technical drawing (not to scale) with a schematic of the rear and forward field of view (FOV) of the camera attached to 7m shark by a 1.8m tether in relation to the shark's swimming speed. The FOV of a camera towed by shark swimming at 0.5 m.s$^{-1}$ is represented in blue while the FOV when swimming at 2 m.s$^{-1}$ is highlighted in red.

behaviours revealed using the duty-cycled video with the continuous video data (collected in 2018), to understand what may have been missed in the duty-cycled data.

## Satellite tracking data

Argos System location data were received during the study period, and were filtered to retain location classes (decreasing spatial accuracy) 3, 2, 1, A and B [66]. No location data were received during the camera attachment period for sharks tagged in 2018 as they were entirely subsurface for their respective deployment periods. Frequent location data were received for

sharks 4, 5 and 6 (in 2019), however, during the camera attachment period. Feeding events and the presence of conspecifics observed in the video data were geolocated using the nearest Argos System locations received within 10 minutes of each event for these three sharks.

## Data analysis

All video data were observed in VLC Media Player (Version 3.0.8). Initial assessment of all data was undertaken by viewing each 30-second video file at twice the native speed to ascertain information on the following: (i) ambient lighting conditions (more than 50% of video data gathered in dark conditions, termed 'blackout', separated into day and night) and (ii) seabed presence and habitat type (whether seabed was visible in more than 50% of data and dominant habitat type). Seabed habitats were classified following the European Nature Information System (EUNIS; https://eunis.eea.europa.eu) habitat classification system, and also included surface and mid-water swimming where neither the seabed nor surface were visible (S1 Table). The frequency and extent of other behaviours, including feeding behaviour (when the shark's mouths was open for more than 50% of the video duration, i.e. >15 seconds), presence of conspecifics and other species were recorded, and species were identified to the finest possible taxonomic level. Where necessary, video data were watched at native or slowed to half speed to ascertain behaviour changes or other events.

To investigate the initial responses of basking sharks to tagging, Tail Beat Frequency (TBF) was calculated as the number of lateral undulatory movements visible in the video data. Using data collected in 2018, where video recording was continuous, TBF was estimated every minute (using 30 second of data), and every five minutes for the 2019 duty cycled data for the first 30 minutes following tagging. TBF was also estimated for each 30-second video sequence of surface feeding behaviour to compare swimming behaviour with previously reported swimming metrics from surface observations [67, 68]. Temperature and depth data collected by the time-depth recorder were analysed in R (version 3.5.3) for the duration that the cameras were recording.

## Results

Basking sharks were instrumented with towed camera tags (2018: n = 2 females, n = 1 male; 2019: n = 2 males, n = 1 unidentified sex) ranging between 5 and 7 m in length. Tags were attached for a mean duration of 63.9 h ± 51.9 s.d. (range 7.8 to 111.1 h; 2018 mean 16.8 h ± 7.8 s.d.; 2019 mean 111.1 h ± 1.5 s.d.). Cameras recorded a cumulative 122.9 h of video data (mean 20.5 h ± 13.1 s.d., range 8.1 to 42.3 h), of which duty-cycled video represented a cumulative 20.8 h of data (mean: 3.5 h ± 3.1 s.d., range 0.8 to 8.5 h). All three rearward facing cameras failed to gather data in 2018 but were successful in 2019. On average (mean), 37% of video data (range 24% to 48%) were gathered at night, or while sharks were swimming at depths, when ambient light conditions were insufficient to observe the shark or its surrounding habitat (S2 Table).

### Tagging response and recovery

The behavioural response to tagging was assessed using video and depth data gathered in the first 30 minutes of tag deployment. For sharks tagged in 2018, when cameras recorded continuously, raw video data were used. Sharks rapidly descended to the seabed following tagging to a mean depth of 28 m (± 20.2 s.d.; range 17.4 to 64 m) apart from shark 4 that descended to 2 m depth before returning to the surface to forage. After tagging, shark 1 swam to deeper waters remaining close to the seabed at least until 50 to 60 m depth below which ambient light conditions were too poor to validate whether the shark was near the seabed. Tail beat frequency and

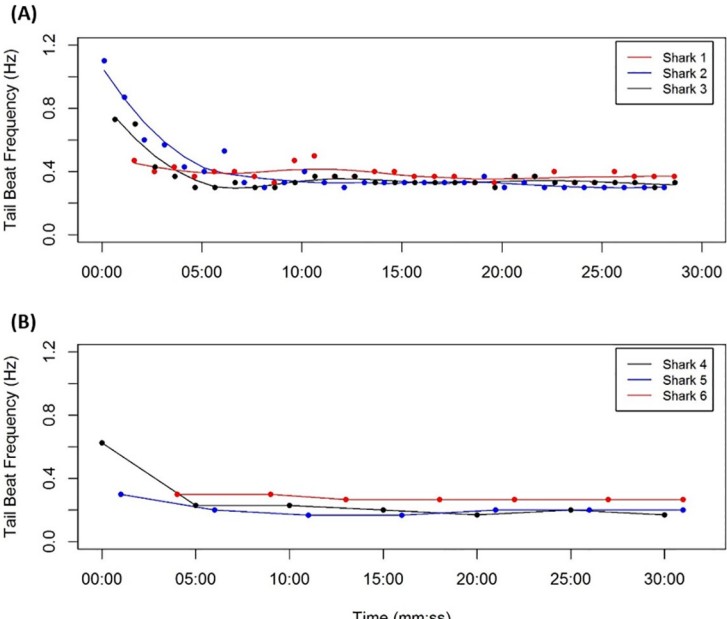

**Fig 2. Response to tagging.** (**A**) Tail beat frequency in response to tagging calculated from 30 second segments of video data taken every minute for first 30 minutes of video data (uses continuously recorded data gathered in 2018 only). (**B**) Tail beat frequency calculated for each 30 second recording period every 5 minutes (for 30 minutes) for duty cycled camera tags deployed in 2019. Local order regression smoothing (colour lines; 0.5 span).

rate of turning indicated that there was considerable variation in the response of individual sharks to tagging (Fig 2). Shark 2 displayed the highest TBF within the first 5 minutes following tagging (1.1 Hz) before stabilising at 0.3 Hz. Shark 1 maintained the highest TBF over the 30 minute period (0.4 Hz) whereas shark 5 revealed the weakest response (0.2 Hz).

## Vertical movement

Depth use varied between sharks, and between years (Fig 3). In 2018, sharks 1, 2 and 3 spent less than 1% of time swimming within a metre of the surface (0.15%, 0.38% and 0.13% of time respectively), while in 2019, sharks 4, 5 and 6 spent longer at the surface (49%, 30% and 15% respectively). Sharks tagged in 2018 swam at greater depth than those tagged in 2019 (mean in 2018 25.6 m ± 13.6, versus mean in 2019 13.6 m ± 3.5), with a maximum depth of 131 m recorded by shark 1. Sharks 1, 2 and 3 were in proximity to the seabed (seabed visible within video; e.g. Fig 3A–3C) for nearly half of the camera operation period (57%, 47% and 42% respectively of the duty cycled data, vs 61%, 48% and 48% respectively in the continuous data). When restricted to daylight hours, during which the vertical position of the shark in the water column could be characterised, sharks 1 to 3 were in proximity to the seabed 57%, 81% and 88% of camera recording time (in the duty cycled data, similar to the continuous data 61%, 82% and 92% respectively, Fig 3A–3C). Shark 1 made repeated short ascents to the surface before returning to 50 to 60 m depth where no behaviour could be inferred from the video data due to insufficient ambient light, whereas sharks 2 and 3 remained predominantly within the upper 30 m of the water column. Comparatively, sharks 4, 5 and 6 spent less than 20% of their camera recording duration near to the seafloor. These three sharks switched between intermittent surface swimming to repeated dives between the surface and the seafloor ("yo-yo" dives), interspersed with excursions to the seafloor (Fig 3D and 3E).

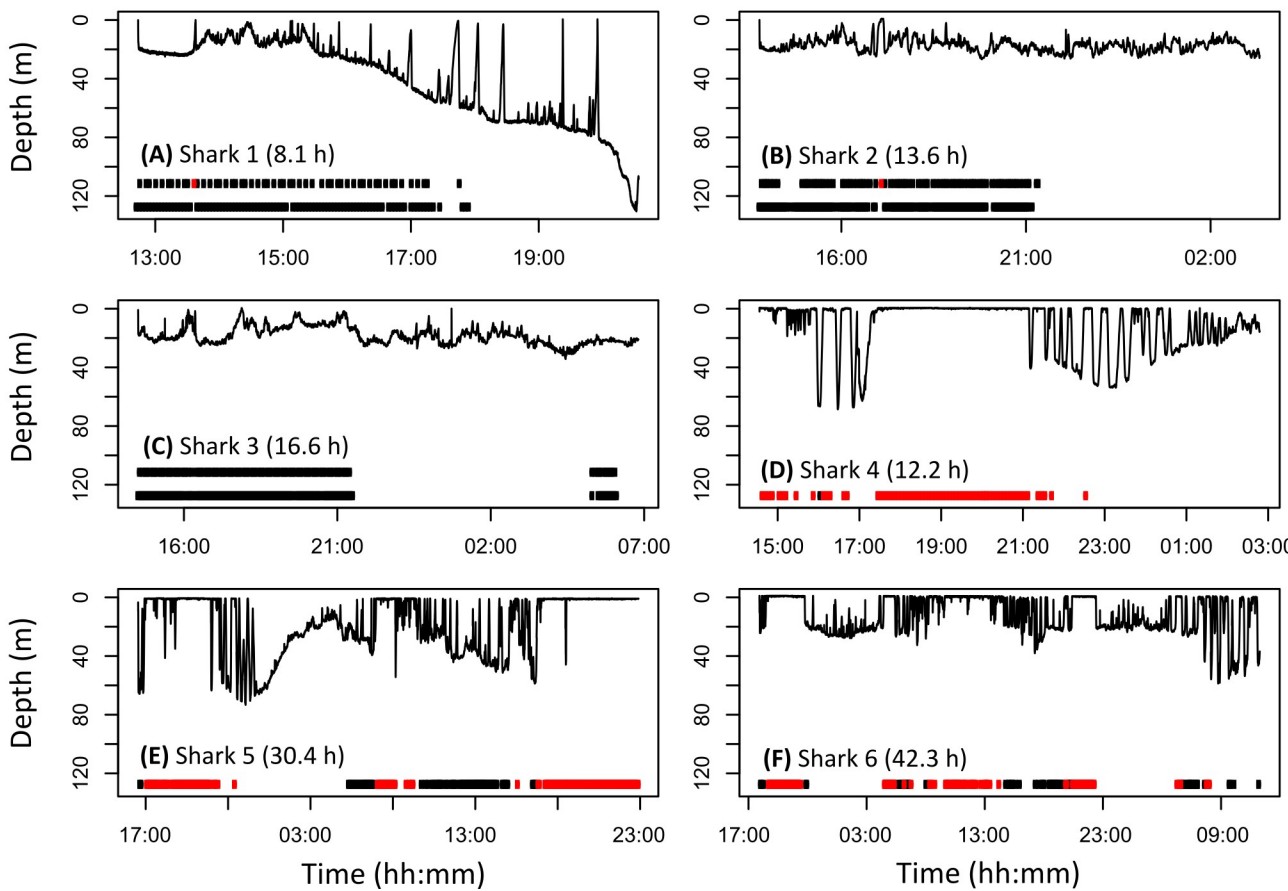

**Fig 3. Depth use behaviour of basking sharks gathered from towed camera tags. (A)** Shark 1, **(B)** shark 2, and **(C)** shark 3 tagged in 2018, and **(D)** shark 4, **(E)** shark 5 and **(F)** shark 6 tagged in 2019. Periods of seabed (black bars) or surface swimming (red bars) when ambient light permitted habitat characterisation, where the top bars in A-C represent depth use inferred from duty-cycled video data, and lower bars continuous data. Tracking duration (h) shown in parentheses.

## Seabed habitat characterisation

Shark 1 spent 36% of daylight hours at depth where insufficient light levels prevented seabed habitat characterisation. When in sufficiently lit conditions, shark 1 was observed over three dominant seabed habitat types: i) infralittoral rock that includes bedrock and boulders with seaweed and kelp communities (EUNIS habitat code A3, 19% of time, S1 Table), ii) circalittoral rock including habitat dominated by macrofaunal communities such as echinoderms (A4, 19%), and iii) sublittoral sediments composed of sand, pebbles and mixed sediment (A5, 18%) (Fig 4A). Sharks 2 and 3 were observed predominantly over kelp, rock and seaweed habitat (A3, 34% of the recording duration each), and 12% and 6% of time over mixed sediment habitat (A5) respectively (Fig 4B and 4C). Shark 2 spent the most time (10%) of all three sharks tagged in 2018 swimming in the mid-water column. Continuous video data for the three sharks tagged in 2018 revealed similar habitat association, with less than 3% variation in habitat type compared to duty-cycled data (S1 Fig). Shark 4 tagged was only observed in proximity to the seabed on a single video recording over rocky substrate (A4) and spent 65% of the recording time at the surface (Fig 4D). Shark 5 swam over the largest range of seabed habitats (Fig 4E) and was associated most frequently with circalittoral habitats (A4, 12%). Shark 6 spent

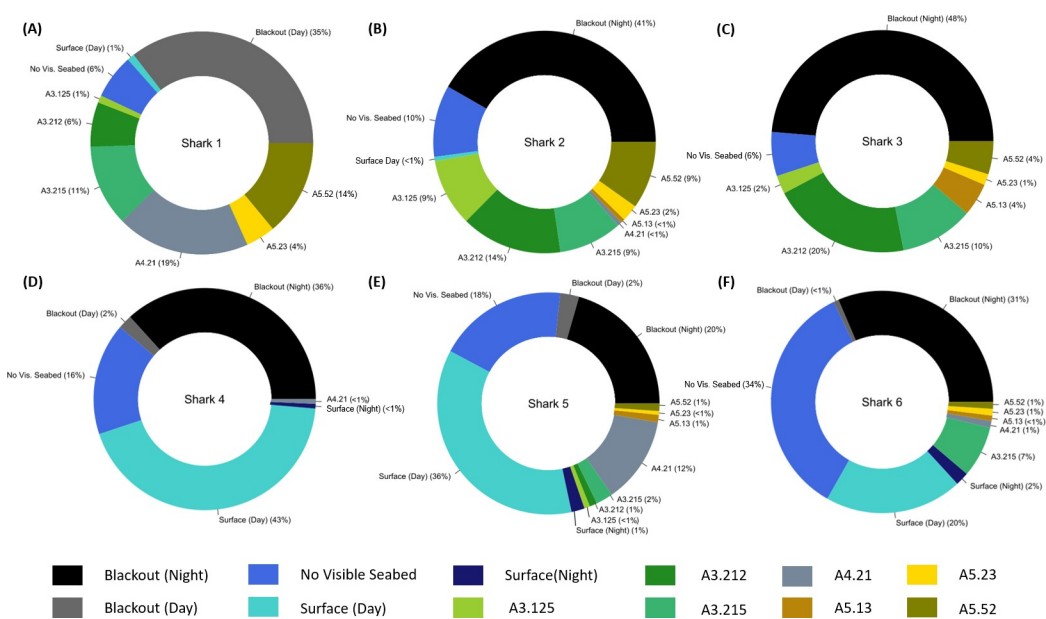

**Fig 4. Habitat association by basking sharks.** Proportion of time sharks 1–6 (**A to F**) spent swimming over different habitat types characterised by their European Nature Information System (EUNIS) code (A3.125 mixed kelp with opportunistic red seaweed on sand-covered infralittoral rock, A3.212 *Laminaria hyperborea* on tide-swept infralittoral rock, A3.215 Dense foliose red seaweeds on silty moderately exposed infralittoral rock, A4.21 Echinoderms and crustose communities on circalittoral rock, A5.13 Infralittoral coarse sediment, A5.23 Infralittoral fine sand, A5.52 Kelp and seaweed communities on sublittoral sediment). Includes proportion of time sharks spent in the water column (No Vis. Seabed), or when habitat could not be characterised owing to poor light conditions from deep diving (Blackout Day) or recording at night (Blackout Night).

the most time of all sharks swimming mid-water (34% of recording duration), and predominantly associated with infralittoral rock dominated by seaweed (A3.215, 7%) (Fig 4F).

## Surface foraging behaviour

None of the three sharks tagged in 2018 were observed feeding. By contrast, in 2019, a cumulative 120 minutes of feeding behaviour at the surface was captured (Fig 5). No sharks were observed feeding sub-surface. The time observed foraging varied between sharks, ranging from 23 to 51 minutes over 1.2 and 4.2 h time periods respectively (period between first and last video revealing foraging). Sharks 4, 5 and 6 were feeding for 35%, 26% and 21% of the camera operation duration respectively. Foraging sharks were usually within 1 m of the surface (average depth ± sd; shark 4: 0.3 m ± 0.2; shark 5: 0.9 ± 0.1; shark 6: 0.6 m ± 0.6) and feeding began before astronomical sunrise (earliest foraging behaviour recorded 04:40 BST), and continued for at least half an hour following astronomical sunset (latest foraging behaviour recorded 22:30 BST). Ambient light became too poor to identify feeding behaviour after 22:30 BST, however, based on time-depth recorder data, shark 5 remained at the surface until 22:55 BST before swimming to depth, so may have foraged during darkness. Sharks 4, 5 and 6 initiated feeding behaviour within 30 minutes of the camera deployment, with shark 4 returning to the surface in less than 20 seconds following tagging and was observed foraging in the first video segment 4 minutes into the deployment. The longest feeding interval was 3 h for shark 4, 3.1 h for shark 5 and 1.5 h for shark 6. Sharks were recorded swallowing 16 times throughout feeding events (3, 6 and 7 times for sharks 4 to 6 respectively). Feeding behaviour represented three quarters of the duty-cycled video data gathered at the surface. If sharks allocated this

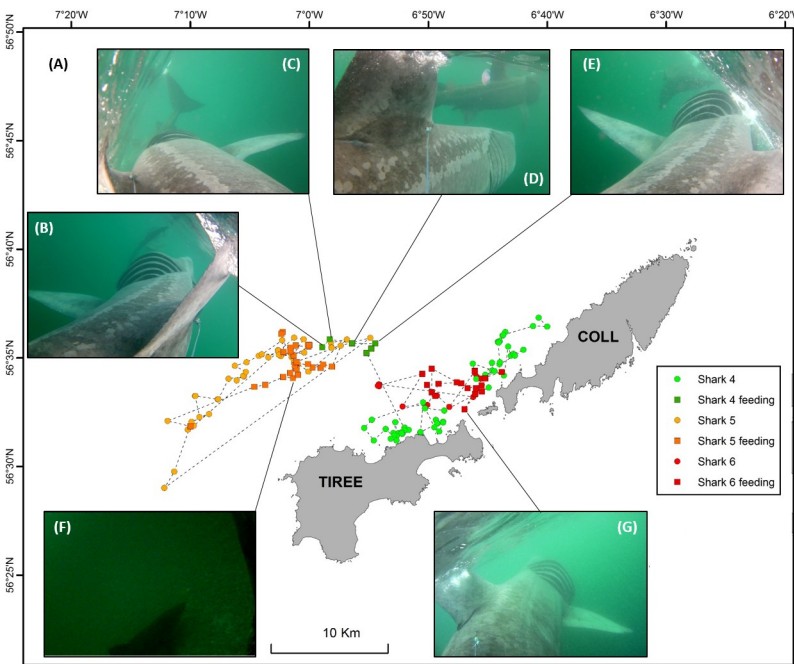

**Fig 5. Basking shark satellite tracking during camera tag attachment. (A)** Satellite tracking locations for shark 4 (green), shark 5 (orange) and shark 6 (red) during camera tag attachment periods in 2019. Location of feeding events for each shark (squares in corresponding colours). Selected behaviours are highlighted with given locations **(B-G).** In **(B)** to **(E)** shark 4 is seen feeding at the surface following a second male shark. Shark 5 camera captures a second shark swimming slowly over the seabed in **(F)** as it swims towards the surface. **(G)** Shark 6 feeding at the surface.

proportion of time to feeding when swimming within 1 m of the surface, sharks 4, 5 and 6 would have fed for 4.6, 7.9 and 11.3 h of their camera deployment duration respectively. TBF of sharks while feeding was 0.20 Hz ± 0.02 for shark 6, 0.21 Hz ± 0.02 for shark 5, and 0.25 Hz ± 0.02 for shark 4. Sharks maintained the same TBF regardless of whether they were foraging or not (Wilcoxon signed rank test V = 853, p = 0.49, foraging 0.22 Hz ± 0.03 versus mouth closed 0.22 Hz ± 0.04), or when being in proximity to the seabed (TBF 0.24 Hz ± 0.08, range 0.17 Hz to 0.38 Hz) compared to the surface (0.22 Hz ± 0.02, range 0.19 Hz to 0.25 Hz). The later comparison could not be tested statistically as only two sharks (sharks 5 and 6) were recorded both at the surface and in proximity to the seafloor.

## Conspecifics

Five towed camera tags recorded other basking sharks on eleven separate occasions, of which six were during surface foraging events (sharks 4 to 6), with the longest interaction lasting 2 h 15 minutes during which shark 4 was following a feeding male shark (Fig 5B–5E and Fig 6E). Sharks maintained similar TBF swimming at the surface regardless of whether in the presence of conspecifics (0.23 Hz ± 0.02), feeding (0.22 Hz) or not (0.22 Hz ± 0.04). The camera tags did not directly reveal mating but did record some courtship-like behaviours and early morning group behaviour. Shark 5 recorded another shark foraging ~15 m away at the surface (Fig 5D), which then stopped feeding and closely approached the tagged shark (within 2 m), switching from its left side to its right by swimming underneath shark 5 before swimming alongside it. Shark 4 followed a second shark very closely (<1 m) at speed (TBF 0.33 Hz) at 60 m depth. The other shark was then observed approaching the caudal fin of shark 4 perpendicularly within 1 m, turned to swim beneath it, and followed shark 4 from below. In the subsequent

video recording, recorded five minutes later, shark 4 was observed close following the second shark nose-to-tail at speed at 29 m depth. A second shark was observed in the continuous video data of shark 1 swimming ~ 20 m ahead and above the tagged sharked at 49 m depth. The second shark was only visible for < 5 secs due to poor visibility at depth. TBF could not be measured owing to instability of the camera. Group behaviour was also observed; shark 3 was observed swimming slowly (TBF: 0.17 Hz, slower than average swimming speed 0.22 Hz) near to the seabed (18–25 m depth) within 5 m of at least nine other basking sharks in the early morning (05:11 to 06:05 BST 3-Aug-2018). The sharks swam closely enough that they touched body and fins two and three times respectively, and swam both closely next to one another and in an echelon formation. Shark 3 was observed swimming ~1 m directly above a second shark in the same direction for 25 seconds before swimming away and following a third shark nose-to-tail, while a fourth and fifth shark swam over the tagged shark, bumping into the camera. In the continuous video data of shark 3, a three minute clip revealed at least nine individual sharks, with an average of four sharks (± 1.9 sharks) sighted per 3 minute clip between 05:11 to 06:05 compared to two sharks (± 0.9, max 4 sharks) over the same time period for the 30 second duty-cycled clips. This is the first evidence to our knowledge of basking sharks aggregating near the seabed (Fig 6E, Shark 5 was observed making an abrupt 4 minute duration dive from

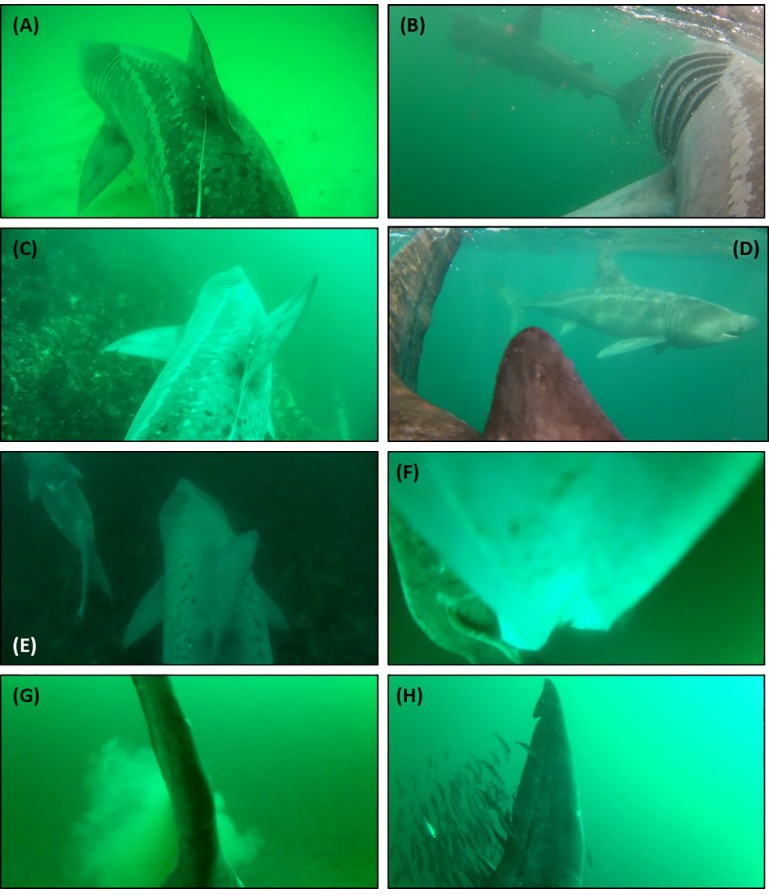

**Fig 6. Observed behaviours from towed camera tags.** **(A)** Shark 3 moving over sandy seabed, **(B)** shark 4 feeding while following a second feeding shark at the surface, **(C)** shark 2 moving over mixed ground with seaweeds **(D)** shark 5 trailed closely by a second shark, **(E)** shark 3 engaged in early morning group behaviour—three sharks visible; instrumented shark (centre) and two other sharks (far left and top right), **(F)** shark 4 rearward camera obscured by plastic sheet, **(G)** shark 5 rearward camera revealing defecation, and **(H)** shark 5 rearward camera revealing small horse mackerel following the tagged shark.

feeding at the surface to the descending to the seabed at 46 m depth at speed where the rear-ward camera captures a second shark swimming slowly over the seabed (Fig 5D) and the tagged shark ascends back to the surface with a tail beat frequency of 0.23 Hz.

## Breaching and defecation

One shark breached clear of the water during recording (Fig 7) at 19:22 BST on the 2-Aug-2018, ascending from 72 m deep at approximately 0.9 m s$^{-1}$ of vertical gain, reaching the surface in 77 seconds The shark exited the water completely before descending to similar depth again (74 m) within 61 seconds at a vertical speed of 1.2 m s$^{-1}$. Five defecation events were recorded in two sharks by the rearward facing cameras, one by shark 4 at the surface, and four times by shark 5 on average 6.1 h apart (± 3.7 s.d., range 1.8 to 8.3 h), twice when near to the seabed (43 and 19 m respectively), and twice mid-water between 10 and 15 m deep (Fig 6G).

## Other species

Video cameras recorded ctenophores, moon jellies (*Aurelia aurita*) and lion's mane (*Cyanea capillata*) jellyfish throughout the water column, species that are consistent for the region [69], with no evidence of sharks either avoiding or eating them. Sharks swam over benthic sessile organisms including sea urchins, brittle and common starfish, sea cucumber, and soft coral (alcyonium). With the exception of shark 6, all sharks had between one and three lamprey attached between the anal fins, as well as behind the dorsal fin in the case of shark 4. All sharks were followed at some point (between 1 and 13% of the video data) by small fish (Fig 6H), likely Atlantic horse mackerel (*Trachurus trachurus*), throughout the water column and near to the seafloor, most commonly over rocky substrate. These fish were cleaning the sharks (or themselves) in over half of the observations (up to 73% (n = 95) for shark 6), and shoaled around the shark's body or trailing their tail (recorded in rearward cameras of sharks 4, 5 and 6) in shoals up to hundreds of fish, although in the majority (80%) of video data there were less than 100 fish. Two instances of antagonistic behaviour occurred, in which sharks 5 and 6 snapped their jaws and made abrupt tight turns of their head in response to shoals of Atlantic horse mackerel appearing to bite the shark caudal fin, followed by three rapid tail beats, dispersing the fish.

## Marine debris

Camera tags also provided incidental evidence of marine debris. Fishing line was observed floating in the water column close to sharks on three separate occasions (shark 5 n = 1, shark 6 n = 2), while a plastic sheet became attached to shark 4's towed camera 35 minutes into the deployment (Fig 6F), obscuring part of the rear view until the end the deployment (11.7 h later).

## Differences between continuous and duty-cycled data

There were few differences in habitat association and behaviours observed in the duty-cycled data compared to the continuous data for sharks 1, 2 and 3. The proportion of time allocated to different habitat types varied by less than 3%. Only three behaviours were missed due to duty-cycling: the sighting of a second shark for 5 seconds by shark 1, two fin touching events during the grouping behaviour at depth by shark 3, and the breach, which was originally identified from the depth profile. The number of sightings of other species remained similar between recording schedules, except for intermittent following of trailing fish which were reported up to four times more frequently in the continuous data (14% compared to 3% of

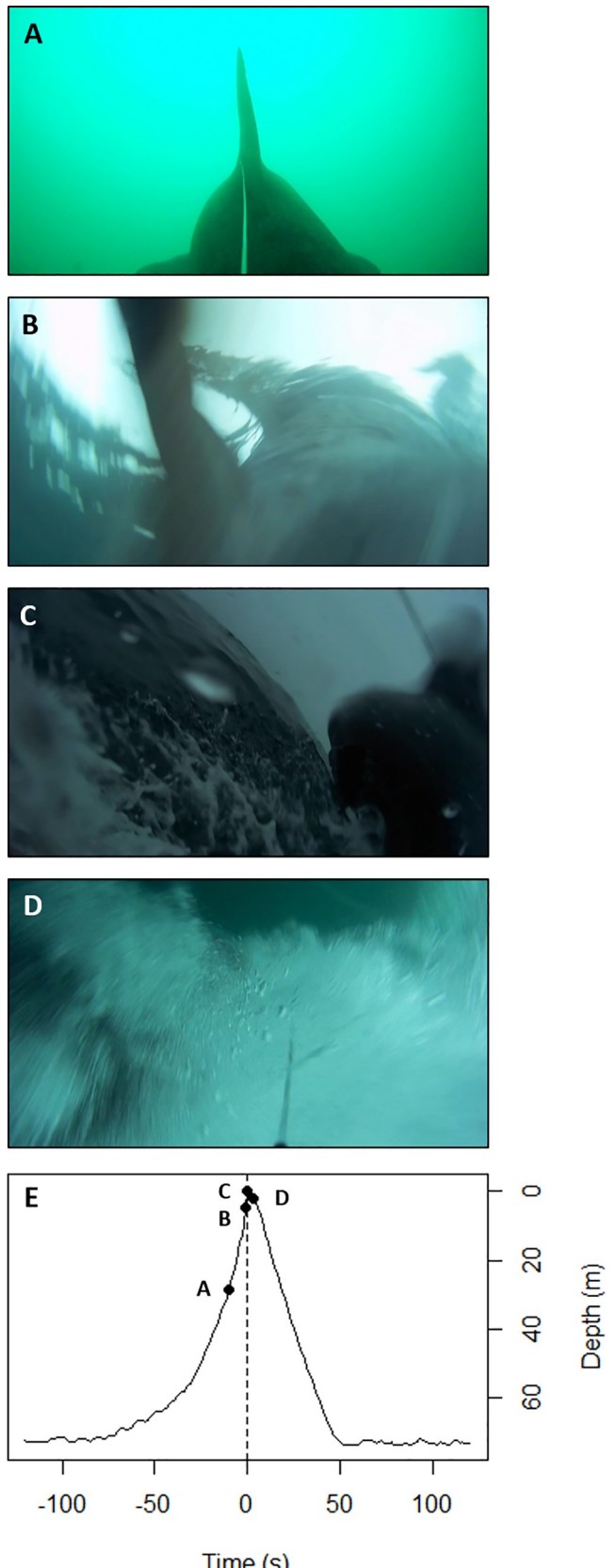

**Fig 7. Basking shark breaching.** Breach recorded by a towed camera tag. **(A)** The shark starts to ascend from 72 m depth at approximately 0.94 m s-1 of vertical gain, reaching the surface (in view, **B**) in 77 seconds The shark can be seen completely out of the water **(C)**, before descending **(D)** to depth again. The timing and depth associated with each image **(A-D)** are identified on the breaching depth profile **(E)**.

clips for shark 3). Most of the basking sharks' noteworthy behaviours such as feeding, trailing or being followed by conspecifics were sustained over periods longer than the four minutes 30 seconds period between video segments. The sub-sampled video data revealed the grouping behaviour over the same overall time scale as the continuous data, but less than half as many sharks were recorded on average (2.1 compared to 4.3), and less than half the maximum number sharks (4 compared to 9) were identified during a single video segment.

## Discussion

### Overview

This study represents the first long-term deployment of animal-borne cameras on basking sharks recording a cumulative 123 hours of video data and revealed the first evidence of sub-surface interaction with conspecifics. Cameras provided unprecedented insight into the social and environmental context of basking shark behaviour, which was previously limited to surface observations, as well as their fine scale habitat association at depth. While basking sharks are thought to aggregate in Scottish waters to feed [50, 57], the sharks in this study spent a surprising proportion of the study period in proximity to the seabed (up to 88% of recordings made in daylight hours) where sharks swam predominantly over infralittoral rock. Knowledge of habitat association is valuable for evaluating the effects of human activity upon the species (e.g. fishing, aquaculture, wild harvesting of natural products, civil engineering, military activities). Any future threat assessments of this species should explicitly consider depth behaviour and seabed habitat associations given the time sharks allocated to swimming near to the seafloor, which increases the possible risk of overlap with human activity sub-surface. For example, in New Zealand, the greatest proportion of accidental capture of basking sharks were by trawlers near or on the seafloor [38]. Towed cameras confirmed the presence of marine debris, including fishing gear, highlighting possible risks of ingestion or entanglement [70, 71]; however, the Inner Hebrides experiences some of lowest levels of pollution from marine waste in the UK [72]. The extent of association with the seabed also has implications for conducting population estimates, since these are based primarily on surface observations [73]. The number of sharks in the region may be underestimated owing to the rather limited time spent at the surface or even in the mid-water column.

Although the tracking duration was too short to robustly test diel patterns of depth use, all sharks swam deeper at night than during the day, supporting previous findings of reverse vertical diel migration in basking sharks in inner shelf areas near thermal fronts [47, 56, 74]. Sharks may have followed the vertical diel migration of prey at night and fed at depth [45, 53], however, ambient light conditions were too dark to identify possible foraging behaviour, and no feeding behaviour was captured even when sharks were discernible on the seabed at dawn. Sharks made intermittent oscillatory dives to the seabed between foraging events, typified by shark 4. While these dives are likely to have multiple functions in sharks [75–78], yo-yo type dives have been suggested to play a role in prey searching [45, 76] as an optimal foraging strategy to maximise encounter rate with spatially and temporally distributed prey such as zooplankton [79]. However, feeding behaviour was only observed at the surface and only for three sharks tagged in 2019. Sharks maintained similar tail beat frequencies when swimming at the surface regardless of whether they were feeding or not and were similar to previously reported

boat-based observation [67]. As ram ventilating species, basking sharks may need to maintain a minimum speed to optimise water flow across the gills to meet oxygen requirements [80], possibly compensating for the high drag associated with filter feeding at the surface. Animal-borne cameras can therefore reveal aspects of basking shark feeding ecology, such as digestion, which may be challenging or impossible to infer from conventional tracking technologies. In the present study, although video data collected was duty-cycled, prey swallowing and defecation events were recorded. If combined with feeding rates, local prey concentration and daily energy expenditure, more accurate energetic models could be populated for basking sharks. While it has been suggested that marine mammals and seabirds play a role in enhancing primary productivity and stimulating carbon export [81–83], the biogeochemical role of basking sharks in the oceans has not been explored. The towed cameras revealed that the sharks defecated at the surface as well as on the seabed, possibly enhancing primary production at depth through fertilisation. While basking sharks are thought to aggregate in the region to feed, sharks fed less than might be expected, with none of the sharks tagged in 2018 feeding in over 16 hours of cumulative video data. Owing to the energetic costs associated with filter feeding, which may be up to twice the costs associated with routine activity at similar speeds [84, 85], basking sharks may have not been feeding because of low prey densities in 2018 at which sharks would not achieve net energy gain [86].

Tagged sharks recorded the presence of conspecifics on eleven separate occasions. Conspecifics were predominantly observed at the surface during feeding events either following or leading the tagged shark closely. While close following is likely to provide a hydrodynamic advantage to the following shark by conserving energy associated with drag or possibly help capture prey missed by the leading shark [57], it has also been suggested to play a role in pre-courtship display [57, 87, 88]. In this study, a female shark was observed swimming towards a male shark from a distance, exhibiting nose-to-tail following, stacking behaviour (swimming below the tagged shark) and parallel swimming while shark 5 was not feeding. Intentional swimming was also recorded by Gore et al. [57], however this type of behaviour was more rarely recorded compared to other close following behaviours. While similar social behaviour has been reported in aggregating sharks at the surface [87–89], this study reported the first sub-surface putative courtship displays for basking sharks. At least nine sharks were observed close following and swimming parallel to shark 3 near to the seabed at dawn, presenting the first evidence of basking shark aggregations at depth. Social groups of sharks are thought to be very short term and sporadic, and may play a role in finding breeding partners [65], particularly in usually solitary sharks such as basking sharks that may use summer aggregations as an opportunity to find mates [87], with high relative population densities possibly triggering mating behaviour in schooling species [90]. Since the tagged sharks maintained similar tailbeat frequencies regardless of being in the presence of conspecifics or alone, both at the surface and at depth, interactions with conspecifics may be missed by using conventional tracking technologies. While no mating behaviour was directly recorded, this study reports previously unknown sub-surface putative courtship displays, highlighting the importance of the region to basking sharks.

Cameras provided unprecedented social context to rare behaviours such as two fast burst chasing events at depth associated with sharks 4 and 5 closely following, or being followed, by a second shark. One chase event lasted over five minutes perhaps suggesting more than an aggressive response. In addition, a complete breach revealing ascent and descent performed by a basking shark was recorded onboard the animal in the present study. Shark 1 was observed to ascend rapidly from the seafloor before clearing the surface and descending back to similar depths. Breaching is observed in a range of species [63, 91–93] and has been associated with communication [94], mate finding [95], predatory ambush [96] and even fun [97]. Because

breaching has been estimated to be relatively energetically demanding [48, 59, 93, 98], it is likely to have a fitness benefit [99]. While the function of breaching remains unclear, it may be related to social behaviour [48, 57, 65]. Breaching has been observed in areas where surface activity, such as close following, has been recorded [57, 87].

With all tagging studies, it is essential to consider the effect of tags on the animal. In this study the response to tagging was measured, with sharks displaying an initial tail beat frequency up to three times higher than the relative baseline frequency. Within 15 minutes (mean 10 minutes), tail beat declined and stabilised, remaining similar throughout the camera operation period. To achieve minimum burden of animal-borne cameras, a trade-off exists with device size, and battery life and/or storage. Several methods have been designed to extend the deployment duration and likelihood of capturing behaviours of interest such as sampling triggered by other sensors (such as depth recorders [100] and accelerometers [19], or selecting sampling rates that are biologically relevant to the tagged species. Hooker *et al.* [100] suggested that when the temporal quantification of a behaviour is important, the sampling rate must be more frequent than the duration of the event, while if investigating the frequency of a behaviour, duty-cycling must take place at intervals shorter than the duration of the behaviour. Basking sharks are slow swimming (mean speed 1 km.h$^{-1}$ [59]) filter feeders with behaviours such as interactions with conspecifics and foraging lasting several minutes to hours. In the present study, camera tags were duty-cycled, sampling 30 seconds every 5 minutes, which enabled the recordings to be extended by about 100% compared with continuous filming (mean 2018 continuous video data 12.6 h ± 4.2, compared to mean 2019 video data 28.3 h ± 15.2). Sub-sampling the 2018 continuous data to match the duty-cycled 2019 footage revealed little variation in habitat association and suggested that duty cycling meant only few incidents were missed. Events such as feeding, defecation, breaching or short-lived interactions between conspecifics were sufficiently sporadic that they would likely only be captured if recording frequency was sufficiently long. While the duty-cycled video data highlighted a similar frequency of common behaviours to the continuous video data, the sampling frequency and interval chosen here are unlikely to be suitable for more dynamic species, such as ambush predators with burst behaviours, low prey encounter rates and short handling times, like the great white shark. Future deployments of cameras with other sensors, such as accelerometers or oceanographic sensors, would also provide important visual context to tracking data. Cameras could also be used to investigate anthropogenic effects in the marine environment on animals at liberty, by directly identifying threats as well as subsequent behavioural responses, which may not be obvious from other types of biologging sensors [101].

## Supporting information

**S1 Fig. Comparison of habitat association by basking sharks.** Proportion of time sharks 1–3 spent swimming over different habitat types characterised by their European Nature Information System (EUNIS) code (A3.125 mixed kelp with opportunistic red seaweed on sand-covered infralittoral rock, A3.212 *Laminaria hyperborea* on tide-swept infralittoral rock, A3.215 Dense foliose red seaweeds on silty moderately exposed infralittoral rock, A4.21 Echinoderms and crustose communities on circalittoral rock, A5.13 Infralittoral coarse sediment, A5.23 Infralittoral fine sand, A5.52 Kelp and seaweed communities on sublittoral sediment) derived from duty-cycled data (**A-C**) and continuous data (**D-F**). Includes proportion of time sharks spent in the water column (No Vis. Seabed), or when habitat could not be characterised owing to poor light conditions from deep diving (Blackout Day) or recording at night (Blackout Night).
(TIF)

**S1 Table. Summary table of the European Nature Information System (EUNIS) habitat codes used to classify the habitat types used by the basking sharks.**
(DOCX)

**S2 Table. Summary information on deployment of towed camera tags (2018 and 2019), including locations of deployment, tag detachment and retrieval times, attachment and data duration and camera performance.** Statistics for sharks 1 to 3 are derived from sub-sampled video data to match the format of duty cycled tags deployed in 2019.
(DOCX)

## Acknowledgments

The authors wish to thank the boat operators, skippers and crew of the Bold Ranger including J. Fairbairns, R. Darby, I. Malcolm, H. Mansfield and R.A. McCann.

## Author Contributions

**Conceptualization:** Lucy A. Hawkes, Suzanne M. Henderson, Marcus Shirley, Matthew J. Witt.

**Data curation:** Jessica L. Rudd, Christopher Kerry, Lucy A. Hawkes, Suzanne M. Henderson, Matthew J. Witt.

**Formal analysis:** Jessica L. Rudd, Haley R. Dolton.

**Funding acquisition:** Lucy A. Hawkes, Suzanne M. Henderson, Matthew J. Witt.

**Investigation:** Jessica L. Rudd, Haley R. Dolton, Lucy A. Hawkes, Suzanne M. Henderson, Matthew J. Witt.

**Methodology:** Tiago Bartolomeu, Owen M. Exeter, Lucy A. Hawkes, Suzanne M. Henderson, Marcus Shirley, Matthew J. Witt.

**Project administration:** Lucy A. Hawkes, Suzanne M. Henderson, Matthew J. Witt.

**Resources:** Tiago Bartolomeu, Lucy A. Hawkes, Marcus Shirley, Matthew J. Witt.

**Software:** Tiago Bartolomeu, Marcus Shirley.

**Supervision:** Lucy A. Hawkes, Matthew J. Witt.

**Validation:** Matthew J. Witt.

**Visualization:** Jessica L. Rudd.

**Writing – original draft:** Jessica L. Rudd.

**Writing – review & editing:** Jessica L. Rudd, Haley R. Dolton, Owen M. Exeter, Christopher Kerry, Lucy A. Hawkes, Suzanne M. Henderson, Matthew J. Witt.

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
