## [Decision Letter · Decision Letter 0]

7 Apr 2021

PONE-D-21-03869

Basking shark sub-surface behaviour revealed by animal-towed cameras

PLOS ONE

Dear Dr. Rudd,

Thank you for submitting your manuscript to PLOS ONE. After careful consideration, we feel that it has merit but does not fully meet PLOS ONE’s publication criteria as it currently stands. Therefore, we invite you to submit a revised version of the manuscript that addresses the points raised during the review process.

In addition to minor editorial comments, please address these four requests from the reviewers:

Subsampling: Given the relative paucity of studies using cameras, it might be worth to compare your 2018 continuous records to your 2018 subsampled records to match the 2019 duty cycle.  Can you please provide some summary statistics to compare the results from the analyses of the data at different temporal resolutions.  As one of the reviewers stated, this analysis would greatly enhance the value and applicability of this paper, guiding subsequent studies with animal-bourne cameras.

Interpolation: Can you use the camera to interpolate behaviors during the “black-out” period?  This could be a great benefit of using cameras over accelerometers/depth recorders alone.

Behavioral States:  In L173, the ms states "[...] (TBF) was used as a measure of behavioural state". One reviewer noted that it is not clear to me to what behavioural states the authors are refereeing to.  Please define the behavioral states are  how they were classified.  You may want to add a table to the supplementary materials.

Figures:  The figures should stand alone. In figure 4, for instance, the information of the EUNIS seabed codes should be included in the figure legend. 

We look forward to receiving your revised manuscript.

Kind regards,

David Hyrenbach, Ph.D.

Academic Editor

PLOS ONE

Journal Requirements:

1. Please ensure that your manuscript meets PLOS ONE's style requirements, including those for file naming. The PLOS ONE style templates can be found at
https://journals.plos.org/plosone/s/file?id=wjVg/PLOSOne_formatting_sample_main_body.pdf andhttps://journals.plos.org/plosone/s/file?id=ba62 /PLOSOne_formatting_sample_title_authors_affiliations.pdf

'The authors have declared that no competing interests exist.'

We note that one or more of the authors are employed by a commercial company: NatureScot.

Additional Editor Comments (if provided):

Reviewers' comments:

Reviewer's Responses to Questions

**Comments to the Author**

1. Is the manuscript technically sound, and do the data support the conclusions?

Reviewer #1: Yes

Reviewer #2: Yes

2. Has the statistical analysis been performed appropriately and rigorously? 

Reviewer #1: Yes

Reviewer #2: Yes

3. Have the authors made all data underlying the findings in their manuscript fully available?

Reviewer #1: Yes

Reviewer #2: Yes

4. Is the manuscript presented in an intelligible fashion and written in standard English?

Reviewer #1: Yes

Reviewer #2: Yes

5. Review Comments to the Author

Reviewer #1: Overview: The authors present the first fine-scale camera biologging study on basking sharks and highlight video evidence of several important behaviors. I was impressed with the samples design, coverage, and analysis. While I would have gravitated towards some statistical comparisons, I appreciate the authors focusing on the observations they recorded and describing the unique behaviors they recorded. I have no qualms about suggesting this paper should be accepted and published. I have included some notes that I think are worth considering but are in no ways necessary for publication.

Notes:

Often a concern in biologging is subsampling. Given the relative paucity of studies using cameras, it might be worthwhile to compare your 2018 continuous records to your 2018 subsampled records to match the 2019 duty cycle. In particular, I would focus on the ability to provide representative summary statistics as well as any potential quantitative comparisons between dive profiles. This would have a high value added effect for the field of biologging and push the study (or subsequent publications) into an area with utility for biologging as a whole.

Can you use the camera to interpolate behaviors in the “black-out” period? I would think that this a real boon to the use of cameras over accelerometers/depth recorders alone. In particular, we know that basking sharks can exhibit some strong diel vertical migratory behavior, especially in UK waters, so this might be particularly useful in this region.

Minor corrections:

Line 19: Reads: “Basking sharks second large shark species in the world” maybe change to “Basking sharks, the second largest….”

-- Dr. Zach Siders

Reviewer #2: The authors tagged six basking sharks with novel towed camera tags. Although the number of tracked sharks is relatively low, their data analysis and findings warrant publication in PLoS One. I enjoyed reading the manuscript, which is very well written and organised. I do not have any major comment. Minor comments below.

#1 L67-70. These two sentences are a bit repetitive. The first says "Animal-borne cameras can also document [...]" and the second "Cameras may also identify possible [...]." Perhaps the authors could merge both in a succinct way?

#2 L173. Methods are quite well explained, except in L173 where it says "[...] (TBF) was used as a measure of behavioural state". It is not clear to me to what behavioural states the authors are refereeing to. Cannot seem to find that information. Please define which states are these and how they were classified.

#3 L180. The sentence defining the feeding behaviour should be moved to near the L167, when other behaviours are being referred to.

#4. L253. Figures are generally quite nice. But they should stand alone. The information of the EUNIS seabed codes should be included in the figure too, as a legend. Referring that information to Table S1 is not good enough.

Figure 7 is amazing.

I have nothing to add to the Discussion. It is quite thorough and all the main findings of the manuscript are discussed. The cited literature is quite exhaustive too.

6. PLOS authors have the option to publish the peer review history of their article (what does this mean?). If published, this will include your full peer review and any attached files.

Reviewer #1: **Yes: **Zachary Siders

Reviewer #2: **Yes: **Nuno Queiroz

---

## [Author Response · Author response to Decision Letter 0]

25 May 2021

Reviewer Comments 1:

1) Given the relative paucity of studies using cameras, it might be worthwhile to compare your 2018 continuous records to your 2018 subsampled records to match the 2019 duty cycle. In particular, I would focus on the ability to provide representative summary statistics as well as any potential quantitative comparisons between dive profiles. This would have a high value added effect for the field of biologging and push the study (or subsequent publications) into an area with utility for biologging as a whole.

We have now done this, and added it to the revised manuscript, see track changes through the methods (line 159), results (lines 243-248, line 273, and line 424-438). We’ve also added some discussion about the differences at line 556-575.

2) Can you use the camera to interpolate behaviors in the “black-out” period? I would think that this a real boon to the use of cameras over accelerometers/depth recorders alone. In particular, we know that basking sharks can exhibit some strong diel vertical migratory behavior, especially in UK waters, so this might be particularly useful in this region.

Unfortunately, the camera did not have a lighting system (which would have drawn on the battery, reduced the recording duration and likely introduce behavioural modification in either the tagged animals or conspecifics), so at depth the footage was totally blacked out. We’ve tried to make this clearer by adding a statement about a lack of lighting to the methods. Thus, no behaviour can be inferred from this video footage except to validate that the sharks have swam to depth and deeper than the capabilities / light sensitivity of the camera, which would be identified from time-depth recorders, or that it is night time.

3) Minor corrections:

Line 19: Reads: “Basking sharks second large shark species in the world” maybe change to “Basking sharks, the second largest….”

Changes made as suggested

Reviewer 2 Comments:

1) L67-70. These two sentences are a bit repetitive. The first says "Animal-borne cameras can also document [...]" and the second "Cameras may also identify possible [...]." Perhaps the authors could merge both in a succinct way?

Sentences now merged to reduce repetition.

2) L173. Methods are quite well explained, except in L173 where it says "[...] (TBF) was used as a measure of behavioural state". It is not clear to me to what behavioural states the authors are refereeing to. Cannot seem to find that information. Please define which states are these and how they were classified.

Deleted “measure if behavioural state” to remove ambiguity of the term.

3) L180. The sentence defining the feeding behaviour should be moved to near the L167, when other behaviours are being referred to.

Changes made as suggested, the sentence was incorporated in L167.

4) L253. Figures are generally quite nice. But they should stand alone. The information of the EUNIS seabed codes should be included in the figure too, as a legend. Referring that information to Table S1 is not good enough.

Descriptions of the EUNIS codes have been included in the legend of figure 4 and no longer references Table S1 for more information.

---

## [Editor Report · Decision Letter 1]

4 Jun 2021

Basking shark sub-surface behaviour revealed by animal-towed cameras

PONE-D-21-03869R1

Dear Dr. Rudd,

We’re pleased to inform you that your manuscript has been judged scientifically suitable for publication and will be formally accepted for publication once it meets all outstanding technical requirements.

Kind regards,

David Hyrenbach, Ph.D.

Academic Editor

PLOS ONE

Additional Editor Comments (optional):

The authors have successfully addressed all the reviewer comments and the ms is ready for publication.
---

## [Editor Report · Acceptance letter]

1 Jul 2021

PONE-D-21-03869R1 

Basking shark sub-surface behaviour revealed by animal-towed cameras 

Dear Dr. Rudd:

I'm pleased to inform you that your manuscript has been deemed suitable for publication in PLOS ONE. Congratulations! Your manuscript is now with our production department. 

Kind regards, 

on behalf of

Dr. David Hyrenbach 

Academic Editor

PLOS ONE